# Exploring Trauma and Resilience among NYS COVID-19 Pandemic Survivors

**DOI:** 10.3390/bs12080249

**Published:** 2022-07-23

**Authors:** Kip V. Thompson, Elizabeth Eder-Moreau, Sara Cunningham, Yuki Yamazaki, Hang-Yi Chen

**Affiliations:** 1New York State Psychiatric Institute, Columbia University Irving Medical Center, 1501 Riverside Drive, New York, NY 10032, USA; 2Division of Psychological and Educational Services, Fordham University, New York, NY 10023, USA; eedermoreau@fordham.edu (E.E.-M.); scunningham16@fordham.edu (S.C.); yyamazaki@fordham.edu (Y.Y.); hchen129@fordham.edu (H.-Y.C.)

**Keywords:** COVID-19, trauma, resilience, telehealth, qualitative data analysis, supervision, training

## Abstract

The New York State (NYS) Office of Mental Health created the NYS COVID-19 Emotional Support Helpline and enlisted graduate students to provide phone-based emotional support initially to the NYS community. This NYS-funded initiative transformed into providing psychosocial support for callers across the United States. Four NYS doctoral students acted as the helpline agents and received 251 individual calls from May–August 2020. The agents documented the calls with clinical notes which cannot be traced back to specific callers. The purpose of this retrospective qualitative study was to explore the themes that emerged from the calls to give voice to the trauma that callers were reporting during the early phases of the pandemic, and the resilience they demonstrated as they engaged with the Helpline. The agents’ clinical transcripts were converted into codes using a critical-constructivist grounded theory approach with the NVIVO qualitative data analysis software. A second research team audited the initial codes for construct clarity. Emergent themes detailed the unique traumas that helpline callers divulged, how the agents provided support, and the callers’ capacities for resilience. Recommendations are suggested to inform clinicians working with pandemic survivors, to offer guidance on providing distance or virtual interventions as well as to enhance policymakers’ understanding of addressing mental health needs across populations served via the NYS COVID-19 Emotional Support Helpline.

## 1. Introduction

### 1.1. Exploring Trauma and Resilience among NYS COVID-19 Pandemic Survivors

The novel coronavirus COVID-19 first gained national attention in the United States in late 2019. Most common COVID-19 symptoms include fever, dry cough, and tiredness, while less common symptoms include a loss of taste or smell, aches and pains, and sore throat [1]. Individuals most negatively impacted by this virus may experience difficulties breathing, chest pain, loss of speech or movement, and, without proper and prompt treatment, death. By early March 2020, COVID-19 had created a global pandemic that, as of the writing of this article, has claimed over 6.2 million deaths worldwide, over 1 million of which have been cases in the United States [1,2].

Beyond the biomedical impact of this virus, the psychosocial consequences of the global pandemic include anxieties due to fears of becoming infected and/or losing loved ones, elevated family and occupational stress, socioeconomic upheaval due to losing one’s employment, and increased isolation [3]. These novel stressors likely contribute to clinically significant levels of psychological and posttraumatic stress disorder (PTSD) symptoms, and the diminishment of resources designed to ameliorate such symptoms [4]. Indeed, the COVID-19 global pandemic has been a catalyst for situation-specific factors that could exceed an individual’s threshold for developing psychopathology [5,6].

### 1.2. Trauma

Those who have survived the COVID-19 global pandemic have been exposed to many difficult emotions, including grief, anxiety, and depression. Many survivors were directly or indirectly exposed to Criterion A traumas of life-threatening events, loss of loved ones, or serious illness. As a result, posttraumatic stress symptoms (PTSS) have emerged among many. Holmes et al. examined the degree to which social workers in the U.S. (*n* = 181) experienced posttraumatic stress between April and May 2020 and found that more than a quarter of this sample met the DSM-5 diagnostic criteria for posttraumatic stress disorder (PTSD) [7]. Bryant-Genevier et al. surveyed 26,174 state, tribal, local, and territorial public health workers to learn the extent of psychopathology among public health workers wrought by the COVID-19 pandemic during the spring of 2021. Their results indicated that 53% of those surveyed reported experiencing some mental health condition in the previous two weeks, and among the four conditions listed, PTSD emerged as the most frequently cited concern above depression, anxiety, and suicidal ideation [8]. 

Social workers and public health workers were not alone in their experience of COVID-19-related trauma—many in the general population who did not have direct exposure to the virus also suffered elevated levels of distress, lost employment and financial status, became isolated, and struggled through abrupt and difficult changes to their routines. Waters et al. cited a national 2020 study that found 13.6% of U.S. citizens surveyed showed psychiatric distress compared to only 3.9% who reported experiencing the same psychopathology in 2018 [2,9]. Perhaps most traumatic for many survivors was the loss of closure through both the inability to say goodbye to dying family members and friends and the powerlessness to hold memorial services for those loved ones [10]. Individuals most at risk for developing PTSD symptoms may be those who had already experienced mental health concerns before the COVID-19 global pandemic and those who became seriously ill themselves or lost others to the virus [10,11].

The COVID-19 global pandemic quickly came to represent a collective trauma, which is defined as the psychological reactions to a traumatic event that negatively impacts an entire society [12]. Collective traumas often change how survivors understand their place within their groups and how they understand their groups’ places within the world; thus, the ramifications of the COVID-19 global pandemic will likely last well beyond the period of immediate risk. Today’s young people will likely remember the trauma their elders experienced from COVID-19 and carry those memories into their twilight years [12]. 

### 1.3. Limited Treatment Options, Uncertainty Abounded

Perhaps nowhere was this collective trauma felt more acutely than the New York City metropolitan area, which had already endured the 9/11 terrorist attacks and was also disproportionately represented in the COVID-19 death toll statistics in the early weeks of the global pandemic [10]. Media outlets spotlighted the overwhelmed NYC hospitals and nursing home facilities in nightly news broadcasts. The World Health Organization’s (WHO) decree that individuals begin social distancing meant that many people could not access their mental health care providers [13], and these same providers protected themselves from risk by discontinuing in-person treatment. Prior to Spring 2020, there were multiple barriers to the proliferation of telepsychology to breach the gaps in mental health treatment that a pandemic of this type caused. Some of these obstacles included trepidation among many psychologists to adopt telepsychology into their clinical practices, a lack of self-efficacy and training regarding telepsychology among some psychologists, and governmental regulations that required patients to live in Medicare-designated rural areas for mental health providers to receive reimbursement from Medicare [14]. This meant that at the moment when many individuals needed mental health treatment the most, the resources became the most challenging to access [15]. 

### 1.4. OMH Helpline

In early April 2020, the New York State (NYS) Office of Mental Health (OMH) announced the creation of the OMH Pandemic Stress Response Practicum for Mental Health Graduate Students. This practicum specialized in Disaster Mental Health (DMH) around COVID-19 by providing disaster-focused psychological first aid, initially to first responders, and later to NYS citizens and then to anyone else across the U.S. experiencing stress responses during the COVID-19 outbreak. These graduate students provided telephonic support via the NYS COVID-19 Emotional Support Helpline after completing psychological first aid training online and other web-based trainings about the practicum, the Helpline, and the intricacies of providing emotional support to callers. They were further trained in how to use an interface called *Aunt Bertha,* which was a database for community resources, such as food, housing, education, and legal services, among others, to provide callers with resources and referrals, as needed. The students, who were referred to as agents on the Helpline, further engaged in ongoing didactics on topics specific to DMH (e.g., coping with grief and trauma) and were supervised by OMH professionals during their shifts. They were directed to provide emotional support and empathic listening but to stop short of providing professional psychotherapeutic services. 

Agents were also required to secure supervision from a licensed psychologist outside of the OMH. The first author assumed the supervisory role and the rest of this research team were agents from the same graduate program. From April to August 2020, this team met once weekly on the online platform, Zoom, to discuss their experiences and provide each other with support. It is well documented that peer support from other clinicians is crucial in mitigating trainee stress in challenging circumstances [5]. 

### 1.5. Telephone Interventions

Research before and since the start of the COVID-19 pandemic supports the use of telemental interventions for individuals experiencing crises and those with unique circumstances [16,17]. Madigan et al. proposed that such interventions have shown effectiveness in lessening the common obstacles to accessing treatment for those living in rural areas, those with transportation challenges, those living within communities without expert mental health care services, and those living in developing countries without high resources for individualized mental health services [3]. Madigan et al. also suggested that navigating online telemental resources might pose a challenge for less tech-savvy consumers, which could deter them from receiving much-needed psychological treatment. These myriad realities make accessing mental health services, provided in-person or online, an insurmountable challenge for certain populations [3]. 

A global pandemic demands interventions that can mitigate the disproportionate harm placed on those populations, yet appropriate community-based mental health services are limited [11]. In situations such as these, telephone interventions may be best equipped to stand in the breach [3,18]. Hausman et al. highlighted the Health Buddy intervention which monitored patients through a telephone device [5]. Health Buddy assessed patients’ symptoms while they were at home and submitted those responses to mental health staff so that they could intervene. Kasckow et al. found generally positive response rates from suicidal veterans living with schizophrenia who received the Health Buddy intervention [19].

Landes et al. sought to survey dialectical behavior therapy (DBT) teams in Veteran Administration (VA) hospitals about their experiences providing DBT services via telehealth platforms, including via telephone [20]. More than 20% of those surveyed DBT providers with VA sites using phone coaching reported that they experienced an uptick in the frequency of calls from their patients from August to October 2020 [20]. An even higher percentage of providers in that specific group reported that the content of the calls began to include increased COVID-19 stress, suicidal behavior, and distress from social distancing. These providers reported that the telephonic intervention made coaching more accessible for their patients and enabled providers to help patients address emotional responses to technical difficulties if they arose; 73% of the providers surveyed rated their patient’s acceptability to this telephonic intervention positively. 

The United States was not the only country that offered telephonic support to its citizens at the dawn of the COVID-19 global pandemic. Switzerland already offered a telephone helpline called “Die Dargebotene Hand” before COVID-19 emerged. This is a free nationwide helpline service that allows anonymous Swiss citizens to speak to volunteers trained in addressing mental or social distress. Helpline operators complete call reports including their best approximation of the caller’s gender and age and three problems the callers reported having. Brűlhart and Lalive analyzed the calls to this helpline, comparing those received from February through May in both 2019 and 2020 to assess the psychological strain caused by the COVID-19 global pandemic among the callers [21]. The results indicated there was not a significant uptick in calls received in 2020 from 2019; however, there was an uptick among callers aged 65 and older. Additionally, problem categories reflecting a fear of infection, loneliness, and struggling with everyday life increased between the two years under examination [21]. 

One year later, Brűlhart et al. published an article further investigating the utility of helplines by documenting the growth and content of helpline calls, as well as those COVID-19-specific elements within those calls [22]. These researchers noted that the callers assumed the psychological and time cost of presenting for support without being prompted to do so, and for this reason, their calls could be considered clinical data because they offered an assessment of mental health that was not impacted by other researchers’ designs and framing. This research team collected data from 23 helplines that were established before the COVID-19 global pandemic across European countries, the U.S., China, Hong Kong, Israel, and Lebanon. The results indicated that across all 23 helplines, calls reached their peak volume approximately six weeks after the COVID-19 global pandemic was formally declared by the WHO (i.e., around late April 2020), and this peak exceeded the pre-pandemic helpline call levels by 35%. Fear was listed as the primary motivator for this increase in calls, followed by loneliness [22]. This research team also found that calls about suicidality, violence, and addiction appeared to decrease between 2019 and early 2021. 

Turkington et al. analyzed the potential effects of the COVID-19 global pandemic on the behavior of callers to a national crisis helpline within the Republic of Ireland [15]. They sought to understand which characteristics of caller behavior may have been transformed by this recent pandemic. Turkington and colleagues found that caller behavior was altered because of the COVID-19 global pandemic and that due to this event, the callers tended to make more calls with a longer duration (e.g., a half hour or longer) and fewer calls with a shorter duration (e.g., five minutes or shorter) [15]. Turkington and his team did not have any demographic information available within this project to add to their data analysis. 

### 1.6. Opportunity for Resilience

The goal of such telephonic interventions, such as the NYS COVID-19 Emotional Support Helpline, is to bolster the resilience of callers during challenging times. COVID-19 and the ensuing pandemic created a collective trauma for individuals across the globe that continues to unfold as this article is being written. Ann Masten’s definition of resilience includes two important judgments: there must be a present threat, and the eventual adaptation or developmental outcome must be assessed as decent [23]. The COVID-19 global pandemic meets the first criterion well, and the opportunity for the second judgment is still available to those who seek it out. 

Pfefferbaum et al. defined community resilience as the ability to “take meaningful, deliberate, collective action to remedy the effect of a problem, including the ability to interpret the environment, intervene, and move on” [24] (p. 349). Building on that knowledge base, Shigemoto conducted a study to examine the potential moderating effect between this construct and traumatic symptoms as a result of the COVID-19 global pandemic [25]. The study results showed that individuals presenting with potential COVID-19 symptoms and those reporting elevated community resilience were likely to report high levels of PTSS. These results also indicated a significant positive relationship between perceived community resilience and higher PTSS [25]. 

These findings are consistent with research conducted by Friis-Healy et al. [11], which indicated that in the context of this public health crisis, resilience often accompanies adaptive emotional responses. To provide clinicians with more tools to bolster resilience in their client populations, Friis-Healy et al. provided a resilience-based intervention framework grounded in the “3Cs” approach to understanding the key tenets of resilience in coping with traumatic stress: control, coherence, and connectedness [11]. We adopt this framework to help explain the resilient nature of the NYS COVID-19 Emotional Support Helpline and the caller–agent interaction that arose from this community-based intervention. Friis-Healy et al. defined control as the self-awareness of having the personal assets necessary to respond appropriately in a crisis [11]. NYS citizens and those from around the U.S. who called this helpline were aware that they had a helpline that they could call to gain support, demonstrating adaptive help-seeking (control). Coherence, meanwhile, signifies making meaning out of challenging situations. Callers to the NYS Helpline initiated a mutual process between themselves and the agent, and within that process for many, the caller explained their narrative and the agent helped the caller to make meaning out of it (coherence). Finally, connectedness was defined as the significance of supportive relationships while coping with crises. NYS Helpline callers became temporarily connected to the agents, and the agents sought to (re)connect callers to new and already existing social networks in the callers’ lives (connectedness). 

### 1.7. Current Study

We propose that the circumstances leading to the creation of the NYS COVID-19 Emotional Support Helpline and the OMH Pandemic Stress Response Practicum are an exemplar of trauma and resilience. The practicum provided an opportunity for those in need of emotional support due to a crisis event to receive emotional support from those able to provide it, and for those providers to gain valuable training and clinical hours whilst serving their community. Multiple studies have emerged in the years since this pandemic began that illustrate the resilience that was borne from this shared trauma; however, to our knowledge, no study provides specific examples of those traumatic and resilient moments. Additionally, few studies investigating COVID-19-related helplines were designed to provide specific excerpts to illustrate the unique dynamics that emerge from caller–agent interactions; thus, this qualitative investigation aimed to retrospectively explore the traumas that NYS COVID-19 Emotional Support Helpline callers were reporting and the resilience they demonstrated through the use of the service. We hypothesized that the callers would report experiencing high levels of anxiety and trauma during the first four months of the COVID-19 global pandemic, and that the Helpline agents would respond with emotional support that would bolster the callers’ resilience in the face of this public health crisis. With this in mind, this qualitative research study was designed to answer the following research questions:What were the most common themes that emerged from these NYS Helpline conversations?What kinds of traumas and anxieties did callers report experiencing?How did callers demonstrate resilience on the telephone?

## 2. Materials and Methods

### 2.1. Study Design

The overall study design was an examination of the themes that most frequently emerged from the qualitative data collected from four agents volunteering for the NYS COVID-19 Emotional Support Helpline. The archival data were collected first as clinical notes, consisting of the nature of the calls and outcomes of the calls, during the agent’s work on the Helpline. After obtaining permission from the Fordham University institutional review board and the NYS Nathan Kline Institute for Psychiatric Research to utilize the clinical notes for the purpose of this retrospective study, the clinical notes were converted to transcripts and coded for themes. The primary objective was to understand the lived experiences of the callers and their experiences utilizing a telephone intervention in the first four months of the COVID-19 global pandemic. The validity of this qualitative data was supported by the fact that the agents completed their clinical documentation without the knowledge that these transcripts would become research data. Thus, there was no motivation to under- or over-report any callers’ concerns. Ethical approval for this study was obtained from the Fordham University institutional review board and the NYS Nathan Kline Institute for Psychiatric Research.

### 2.2. Sampling Strategy

Four agents documented all calls received from the NYS COVID-19 Emotional Support Helpline between May and August 2020. The clinical documentation included the general demographics of anonymous individual callers that could be gleaned over the phone, the reason for the individual’s call, what steps the agent took to address that reason, and the outcome of the telephonic interaction. The agents submitted these clinical notes to the first author weekly from May to August 2020. All clinical documents submitted were included in the data analytic strategy and any identifiable information (besides gender) that could be traced back to the callers was removed to ensure confidentiality. The services provided via the NYS COVID-19 Emotional Support Helpline were anonymous and confidential and the research team cannot trace the data back to specific callers. Additionally, the callers’ names were not collected or recorded, further protecting the callers’ confidentiality.

Between these four agents, 251 calls were received; it is unknown how many of these calls were from repeat callers. Based on the agents’ best estimations, 69% of these calls were from female-identified callers (*n* = 174), 29% were from male-identified callers (*n* = 75), and 0.7% were from gender non-conforming or otherwise gender unknown callers (*n* = 2). 

### 2.3. Data Analysis

The clinical documentation was converted into qualitative data using the NVivo 12 software. Using a qualitative coding approach informed by critical-constructivist grounded theory [26], the second, third, and fourth categories included in the clinical documentation were transformed into parent nodes. With these three parent nodes established, the first author divided the clinical documentation into five groupings such that each research team member reviewed and coded 20% of the agent’s clinical documentation for one month of calls; thus, the four agents/doctoral students also served as the initial coders for this project. Every two weeks, the first author continued the division and assignment process until all calls throughout the four months of data collection were reviewed and coded. The research team met virtually once every three weeks to discuss coding issues and suggest new parent and child nodes for the developing coding structure. This process began in March 2021 and concluded in May 2021. 

In May 2021, the research team met to finalize the coding structure and the code definition book. Next, the research team coded and reviewed the clinical documentation once more to ensure that all documents were coded according to the final coding structure. After the second wave of coding was completed, the first author recruited a new team of recent M.S. Ed graduates to audit the codes. In this phase, the first author downloaded the codes into Microsoft Word documents and distributed an equal number of pages into five groupings for each auditing member to review to ensure each entry fit the final code definition book and to recode the entries that did not. The auditors were also responsible for double coding entries that fit more than one coding definition. The auditing process began in early July 2021 and concluded in late August 2021; during this time, the first author met with the auditing team twice to answer questions and discuss discrepancies.

## 3. Results

Data analysis yielded a coding structure that started with three parent nodes that reflected the general flow of all calls received (i.e., reasons for the individual’s call became the Personal Concerns parent node, steps the agent took to address callers’ reasons became the Recommendations parent node, and outcomes of the telephonic interactions became the How Call Ended parent node). From each of these three parent nodes (i.e., Personal Concerns, Recommendations, and How Call Ended) sprang the child nodes that illustrated more specific types of experiences described by the parent nodes. Eventually, six more parent nodes emerged that reflected aspects of most calls that were not part of the predicted flow of these calls (i.e., Cultural Issues, Extreme Distress, Family Concerns, Frequent Callers, Just Called to Check In, and Powerless to Help Caller). These six parent nodes did not produce child nodes. Figure A1 and Figure A2 (Appendix A) illustrate these parent, child, and grandchild nodes. Only those child and grandchild nodes with 15 or more entries were included here; nodes with asterisks indicate those nodes that reflected the flow of most Helpline calls. 

The themes are categorized according to the research questions/hypotheses about the lived experiences of those utilizing the NYS COVID-19 Emotional Support Helpline. We hypothesized that the callers would report elevated levels of trauma and anxiety related to COVID-19 during the calls. We also hypothesized that the support provided by the psychology trainees working on the helpline would foster resilience among the callers; thus, the themes fell into two overarching categories, Trauma and Anxiety, and Resilience (see Figure A3).

We aim to describe the themes reflecting the callers’ trauma and anxiety, and the ways that agents helped the callers demonstrate resilience when faced with extreme distress through description and quotes taken from the agents’ clinical documentation. 

### 3.1. Trauma and Anxiety

The callers had several experiences that they found traumatic as the result of the COVID-19 pandemic, quarantine, and the psychological and emotional consequences of the same. In general, the callers’ trauma responses can be categorized as a heightened sensitivity to stress and negative emotions such as loneliness and guilt. The callers also expressed feeling isolated from others because of quarantine and differing emotional experiences in the pandemic. The present section of the results describes each of these themes in greater detail, while also providing insight into the specific origins of the callers’ distress. 

#### 3.1.1. Extreme Distress

The callers expressed heightened levels of stress to the agents, and their psychological stress responses often manifested as emotional volatility (i.e., crying or yelling at agents) or panic symptoms (i.e., being unable to control fears). The callers also endorsed physical symptoms of panic, including a rapid heart rate, difficulty breathing, and an overall heightened sense of fear and hypervigilance. The following quote from an agent’s clinical documentation provides an example of the stress observed:

“Caller called to relate her frustration seeing people not wearing face coverings. She’s stressed that mask wearing is not mandatory, and [believes] this was caused by the misinformation shared by the government and through the media. Caller said she had called multiple customer service/informational helpline outlets including ones in [her home city] regarding this problem with the hope to change the policy. Caller said she always wanted to be preemptive especially since both caller and her son are immune compromised. She feels paranoid when she goes to the store because many people are not wearing masks or taking precautions. Yesterday, when she saw a small child without a mask in the store, she went to the child’s mother who then later went after her and yelled about it. Caller said there were other occasions where she spoke up and got rejected or offended, so she is now concerned that her ‘big mouth’ will always get her into trouble, and she should just keep things to herself. Caller said she feels bothered that people are so ignorant and feels discouraged by mankind. She couldn’t get rid of the thought that ‘we would be over it if we all comply with it.’ Caller was very expressive, and her speeches were tangential—she switched between topics that did not flow.”

In addition to experiencing fear, frustration, hypervigilance, and irritability as evidenced by the previous quote, the callers also expressed augmented personal concerns due to the COVID-19 global pandemic. These concerns undoubtedly contributed to the heightened stress responses that the callers reported due to the pandemic.

#### 3.1.2. Family Concerns

When the callers expressed a preoccupation with family members having COVID-19, distress in family relationships due to quarantine, or fear about family members’ mental health due to the pandemic, their call was coded as Family Concern. These calls were frequent, given the nature of quarantine in which families found themselves in close quarters for longer periods of time than they had experienced before. This proximity during a stressful time sometimes contributed to heightened interpersonal conflicts. Other times, the forced isolation and distance between family members due to quarantine requirements contributed to stress, sadness, anger, frustration, guilt, or other negative emotions. For example: 

“Caller called to seek support as her entire family had caught COVID-19 which started with her husband. She is dealing with (1) resentment towards her husband for bringing this disease and (2) guilt that she could not hold her newborn baby and other children because of the virus. Had the husband on the phone but he did not speak. The caller teared up when speaking about her stress related to not being able to take good care of her kids.” 

Additionally, many callers were concerned about loved ones who were in the hospital due to COVID-19, while others were fearful for family members’ mental health. One agent’s call log reflected concerns about family members’ mental health in the following quote: “The caller stated that she was worried about her adult son. She reported that he was ‘not sleeping for two weeks, agitated, having very negative thoughts.’ She stated, ‘I don’t know what to do.’”

Compounded within concerns about family members were callers’ personal concerns about their safety and emotional and financial well-being. 

#### 3.1.3. Personal Concerns

Increased personal concerns were another one of the impacts that the agents witnessed while working with the callers. This category of calls describes the callers who discussed fears of contracting COVID-19, the inaccessibility of medical care, loneliness, and employment or financial concerns. 

##### Contracting COVID-19 

Many callers were naturally fearful of contracting COVID-19 and called the helpline for emotional and practical support regarding the same. The callers described fears and panic that they might perish if they caught COVID-19, and some expressed fears about contracting COVID-19 more than once. One agent captured a caller’s concern about contracting COVID-19 due to their immunocompromised status:

“The caller explained that she calls the Helpline a few times a week due to anxiety about getting COVID-19. The caller stated that she would not survive if she got the virus due to a medical condition. She stated that she is extremely careful and cautious but cannot eliminate all risks. The caller described many instances in which she could have been exposed during the past week, despite being in quarantine. She reported that she lives alone with two cats and her closest friend lives an hour away. The caller shared that she has a sore throat and that it is making her worried that she could have COVID-19. She described the reason for her call to be seeking support for coping with the anxiety and fear…”

Some callers noted that they had developed a fear of leaving their homes due to heightened anxiety about contracting the virus. For example, one agent recorded: “Caller worried she had contracted coronavirus by inhaling a draft from outside her door. The caller appears to have developed symptoms of agoraphobia.”

Even when callers were able to leave their homes, they expressed emotional distress related to social distancing and quarantine. 

##### Loneliness

Perhaps one of the most prevalent preoccupations callers voiced was loneliness and isolation caused by quarantine. One agent wrote:

“Caller called saying she was alone and had no one to talk to. Said she was very worried about the virus and had ‘absolutely no one’. When I tried to ask about neighbors/people in her building, the caller became very upset that I would ‘question’ her and then hung up.”

Another agent reported the following concerning a caller’s difficulty being alone: “Expressed feeling overwhelmed and depressed, changes in sleeping, eating, mood, activity. Described having difficulty coping with being alone.”

Although many callers did not live alone such as the caller described above, they still reported increased emotional and interpersonal distress due to feelings of alienation from their communities, friends, and loved ones.

##### Surrounded by Death

Another source of distress unique to specific callers, but frequent nonetheless, was direct and indirect exposure to an overwhelming number of deaths and fatalities due to COVID-19. Callers who lost loved ones to COVID-19 expressed feelings of sadness, anger, frustration, and grief about loved ones dying from COVID-19. For example, one agent’s log described a call where the caller discussed losing more than one family member to COVID-19: “Caller’s cousin has been hospitalized for COVID-19 and recovering from a critical condition. The cousin’s uncle recently passed from COVID.”

Another reads: “The caller shared that she lost her mother from COVID-19 recently and was having difficulty coping. She shared ‘I don’t know what to do.’ She stated that she needed to talk about it.”

COVID-19 mortality rates were overwhelming even for those not directly exposed through the loss of a loved one, as one agent relates in the following note:

“The caller stated that she works for the government processing death tickets and benefits. She was tearful and described feeling overwhelmed with sadness about all of the death tickets she has had to process. She shared that she has to speak with the families of those who passed away and that it has been overwhelming to hear so many sad stories. The caller shared that she loves her job and feels grateful she is able to help the families feel better and provide support to them during such a difficult time, but that the work ‘gets to her’ after a while. The caller also shared that she is finding it difficult to homeschool/teach her children and that she has no energy for it after ‘witnessing death’ all day.”

Compounding each of these personal concerns that were more emotional in nature were practical stressors about employment, as many callers experienced job and financial insecurity due to the pandemic. 

##### Employment Concerns

Concerns surrounding employment centered around workplace safety and financial distress from potential or actual job loss. Callers often discussed feeling unsafe in the workplace when forced to work in person and described feeling vigilant of others’ use of masks. Although callers felt in danger in their workplace because of COVID-19, many were also unable to leave their jobs due to financial concerns. One agent’s note describes a woman who called, stating:

“The caller stated that she was calling to see if what she was experiencing was ‘normal’ or not. She stated that she works in retail and has been feeling frustrated and concerned that customers do not respect her space, wear masks incorrectly, or don’t wear them at all, and that her employer is not enforcing mask-wearing or other safety policies. The caller stated that a co-worker came to work who appeared and reported feeling ill with symptoms that sounded like they could be Coronavirus symptoms. The caller reported that her co-worker asked to go home, but their boss denied the co-worker’s request. The caller stated that this incident made her feel extremely unsafe and worried about her own safety and the safety of her family. She reported the incident to what she thought was the Department of Health, but realized it was the Department of Labor… and since this incident, her boss has been treating her ‘unfairly’ and she feels her job is in jeopardy.”

One agent also described callers who had COVID-19 and their resulting concerns about employment, stating: “Caller called looking for financial support resources because he lost [his] job due to COVID and was hospitalized for COVID and is now struggling to provide for the family. Has two teen children.”

Taken together, it was evident from callers’ reasons for calling that there were several types of personal concerns resulting from the pandemic that also negatively impacted their mental health. 

### 3.2. Resilience

The callers demonstrated resilience during the calls as they worked with the agents to identify prior coping strategies, connect with resources or referrals in their communities, or engage in problem solving despite their challenging circumstances. The following categories captured how agents engaged the callers’ resilience via (1) psychosocial interventions, (2) referrals, and (3) miscellaneous problem solving. 

#### 3.2.1. Psychosocial Interventions 

This node describes instances where the agent provided therapeutic support and services to the caller while on the phone. Examples include validation/support, exploring existing coping skills, teaching new coping skills, and/or psychoeducation. Callers exhibited resilience as they identified the qualities, strategies, and supports they were using to help them cope with their circumstances. The following clinical note captured one caller’s resilience as the agent worked with them to identify the strategies and perspectives they were utilizing to help them cope with the loneliness and isolation they were feeling. The agent recorded:

“Although seemingly lonely, the caller said that it doesn’t take much for him to be happy. He named quite a few coping skills including music, arts (clay, painting, coloring), cooking, and taking walks. I praised the caller for being resilient and resourceful, and reminded him that he might not have control over how others react to him, but he has control over taking care of his own wellbeing through these various coping strategies.”

Like this caller, many callers underestimated how much they were already doing to support their mental health. They benefited from having someone validate their concerns and help them identify ways to improve their moods or circumstances that were within their reach, such as taking a walk. The following quote demonstrates the same:

“I provided emotional validation to the Caller’s feelings of loneliness and fear. I tried to explore her current coping, to which she indicated cleaning, moving furniture, and listening to music have been helpful. When I asked about deep breathing and meditation to manage fear and improve sleep, she said they do not work for her…I also explored her interest in speaking to someone for therapy on a regular basis for emotional support and human connection. Caller was interested in seeing a therapist.”

While not all calls resulted in a referral to therapy, the coping strategies sometimes involved connecting the callers with sources for ongoing support, such as was the case with the above call. The agents also assisted callers to recognize their capacity for resilience by identifying ways they had managed stress in the past or were already positively managing the stress of the COVID-19 global pandemic. For example, one agent recorded an interaction with a caller who identified as a farmer and who was having difficulty coping with the impact of the pandemic on their life, writing: “I listened and provided support and reflected upon resilience he has shown thus far by maintaining the family farm for so long through other national economic crises.”

In sum, it was helpful to the callers when the agents validated their feelings and reminded them of their strengths and areas of their lives in which they were excelling, helping them to redirect their focus away from the stress of the COVID-19 pandemic.

While the agents were often successful at providing resources or referrals to callers for ongoing support, many callers rejected the agents’ suggestions that they speak with a therapist due to cultural stigma or the feeling they did not need that type of support. For example, one agent recorded: “…caller declined referrals for therapy because therapy is for, in his words, crazies.”

The Helpline seemed to offer support to individuals who saw themselves as struggling with their circumstances but not in need of professional help. For some callers, they needed temporary validation, support, and connection from the Helpline during the heightened stress, losses, and traumas brought on by the pandemic, but they held an underlying belief they would get through it with their internal and external resources over time. For other callers, cultural stigma and barriers to accessing mental health care impeded their ability to connect with needed professional support. 

#### 3.2.2. Referrals

Another way the agents provided support was by connecting callers to community resources or referrals. This theme describes instances where the agent made community-based recommendations (e.g., other hotlines, social services, mental health referrals, medical-related referrals, etc.). The ways in which the callers engaged with the agents regarding resources and referrals were unique. For example, some callers utilized the Helpline for specific purposes and asked for referrals directly. The following clinical note demonstrates an example of how some callers engaged with the agents to find appropriate referrals:

“I described the services I would be able to provide the caller with information about (free text therapy platforms, such as Talkspace) or to put the caller in connection with (therapy referrals via Psychology Today or Aunt Bertha). The caller wanted to discuss her options for ongoing individual psychotherapy only. I collected information about the caller’s insurance, zip code, and preferences for a therapist. The caller stated that she was interested in a Black or Latina female therapist who specializes in anxiety and depression. I filtered my search based on all the criteria and found 3 options for the caller. I provided their names, locations, phone numbers, and cost/sliding scale options as listed on Psychology Today. I then provided the caller with 1-888-NYC-WELL in case none of those worked out.”

This caller’s ability to express their preferences for individual therapy and a therapist of color exemplifies how some callers actively participated in seeking support and resources to help themselves and their families cope with stressors brought on by or worsened by the pandemic. 

#### 3.2.3. Problem Solving

Whenever the solutions the agents provided did not fall under the category of psychosocial interventions or referrals, they were coded as Problem Solving. Such interventions included everyday problem solving or teaching callers specific techniques (e.g., deep breathing). The agents recorded caller responses to their interventions, which were generally described as helpful by the callers. 

Due to the nature of quarantine, some callers attempted to use the Helpline as a means for everyday problem solving and to help them with carrying about their otherwise regular activities in the “new normal” of quarantine and isolation. For example, one caller called to voice their frustration about their inability to attend church services because of the pandemic. The following describes the agent’s response: 

“I provided empathy and support. Looked up live streaming Baptist church services. Provided her with a few options for recorded services but could not find any live services for tonight. I also provided the caller with Invisible Hands as she described being in need of groceries and bottled water.”

This quote demonstrates the ways that the agents used information provided by the hotline and the internet to assist the callers with practical matters. Other practical matters callers required assistance with were understanding quarantine restrictions, such as for travel. One agent writes:

“I provided the caller with the DOH Coronavirus Hotline number and then provided clarification about the 14-day quarantine regulation for New York State per the ‘Interim Guidance for Quarantine Restrictions on Travelers Arriving in New York State Following Out of State Travel’ available on the New York State Website.”

Callers also received support from the agents by learning specific breathing techniques and other coping strategies to manage their anxiety and distress due to the pandemic. One agent wrote, “…engaged the caller in the following: 4-4-4 breathing (caller stated that 4-7-8 was too long), belly breathing, visualization exercises, progressive muscle relaxation exercise, mindfulness/grounding activity.”

By providing the callers with these interventions and resources, the agents were able to increase the likelihood that the callers would feel more effective and capable of managing their stress during the pandemic.

### 3.3. Results Summary

Figure A3 was designed to illustrate the traumatic content that emerged from three parent nodes (i.e., Personal Concerns, Extreme Distress, and Family Concerns), and the resilient content that emerged from three child nodes of the Recommendations parent node (i.e., Psychosocial Interventions, Problem Solving, and Referrals). Not every caller demonstrated both trauma and resilience: most conveyed some form of trauma and/or anxiety but not all showed resilience as well; thus, there is no predictive relationship between trauma and resilience implied in this graphic. The graphic is provided to give the reader an idea of how the authors grouped these two constructs according to the qualitative data that emerged.

## 4. Discussion

Through the present study, the researchers sought to retroactively determine the experiences of trauma and resilience of those who called the NYS COVID-19 Emotional Support Helpline. Calls were coded by psychology doctoral graduate students who worked as agents on a COVID-19 helpline as part of a practicum experience required for their doctoral degrees. The coding revealed 9 parent nodes/themes (see Appendix A Figure A1), 39 child nodes/subthemes, and 4 grandchild nodes/subthemes (see Appendix A Figure A2). From these themes and subthemes emerged ample evidence to suggest that the callers experienced symptoms of different posttraumatic stress reactions while also demonstrating resilience with the assistance and support of the agents. 

In addressing the research questions, the most common theme that emerged from the NYS Helpline was the trauma and anxiety expressed by many callers to the agents. These reported experiences of trauma and anxiety took on various forms, coded as one of the multiple parent nodes identified, from extreme distress over contracting the virus to managing anxiety regarding everyday problem solving. Demonstrative of trauma responses in the agents’ notes is the heightened sensitivity to stress and negative emotions the callers presented with (i.e., a caller expressing offense when an agent suggested looking for support from neighbors when the caller reported “having no one”). Continuing to the second research question, the traumas reported by the callers aligned with referenced literature such as callers being inundated with information and a lack of closure regarding death and dying, chronic fear of developing and/or guilt in spreading COVID-19, and profound feelings of isolation due to quarantine and social distancing. Despite this, the callers were able to demonstrate resilience through being able to identify coping skills already in use, reflecting on times they were resilient in the past, and connecting with the agents to engage with opportunities for further support, such as individual therapy. To this latter point, the callers and agents were able to actualize the “3Cs” grounded framework to conceptualize resilience in coping with traumatic stress through control, coherence, and connectedness [11]. 

Other studies examining the impact of COVID-19 on communities across the globe discovered similar depictions of trauma and anxiety to those found in the current study, such as Mboua et al.’s study that reported elevated PTSD symptoms for Cameroon citizens during the pandemic [27]. This study’s findings also parallel others examining resilience amidst the COVID-19 pandemic that suggest that most individuals have the capacity for resilience, particularly when able to connect with others [28,29]. Beyond connecting and sharing their troubles with the agents, this study uniquely demonstrated the specific interventions (e.g., providing psychosocial support, referrals, and engaging in problem solving) that engaged the caller’s resilience and helped them cope with their circumstances.

### Limitations and Future Directions

There are a few limitations to the present study which restrict the generalizability of our findings and should be considered when contemplating the results presented. Perhaps one of the most important is that the authors could not adequately consider the ways that culture may have influenced the results or coding because demographics were not collected from the callers. Prior research indicates individual cultural variables, such as race, and state-level variables, such as state GDP, may impact trauma and resilience in unique ways [25]. A second limitation of the present study is the retroactive nature of the research question. That is, the agents did not begin the calls thinking that they would be coded for qualitative research; instead, they were simply attempting to support the callers. The authors decided to use the agent notes for qualitative analysis after leaving the helpline, such that there was not a specific list of questions that the callers were asked as in other qualitative interview studies. Further, although necessitated by the nature of the pandemic, the telephone has specific weaknesses in relation to qualitative research such as a loss of visual cues from the participants, including body language and facial expressions [30]. In turn, for themes such as emotional volatility, the agents could only base their notes and perceptions of initial calls on the callers’ tone of voice and speech; however, because of the pandemic, this limitation could not have been avoided even if the study had been designed as a prospective analysis. Such limitations could be addressed in future research, as helplines continue to exist and merit further study as telehealth becomes increasingly more popular.

As the country has shifted away from in-person services to include telehealth, it is imperative that researchers continue to assess the effectiveness and efficacy of such interventions. Inherent in such questions is the impact of modifications of the traditional in-person therapeutic space on confidentiality and traditional ethical standards (i.e., confidentiality) [3]. To that end, it is important that graduate psychology and other mental health services programs begin to address training in telehealth to meet the evident need to reach populations who cannot access in-person services [5,31]. Legislators at both the state and federal levels are encouraged to fund community-based resources that provide free, competent, telehealth interventions such as the NYS COVID-19 Emotional Support Helpline, and not just during times of crises. Future helpline developers should consider using social media platforms to advertise these helpful resources. Finally, as a means of considering culture in such interventions, it might be relevant to compare the outcomes of helplines in different countries more systematically while noting caller demographics. 

This research suggests that telephonic community-based resources have the potential to harness the resilience which is innate within individuals. Our communities benefit when mental health professionals, elected leaders, and educators join forces to ensure such resources are readily available in every state, territory, jurisdiction, and country across the globe.

## Data Availability

Data supporting these results can be found by contacting the first author directly.

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
