# Peer review of "Exploring Trauma and Resilience among NYS COVID-19 Pandemic Survivors"

_behavsci, 2022, doi:10.3390/bs12080249_

Round 1

Reviewer 1 Report

This is a well done study of remote support provided to callers who experiences stress, anxiety or who sought information and support in relation to the COVID pandemic. It effectively uses the trauma and resilience model to organize the research analysis. The qualitative methods used are appropriate to the exploratory nature of the study. The methodology is explained well. The findings are consistent with the rapidly emerging research on COVID and mental health and well-being. It is an original contribution to the literature and expands on the role of tele-health and remote support and psychoeducation.

Author Response

Reviewer comment: This is a well done study of remote support provided to callers who experiences stress, anxiety or who sought information and support in relation to the COVID pandemic. It effectively uses the trauma and resilience model to organize the research analysis. The qualitative methods used are appropriate to the exploratory nature of the study. The methodology is explained well. The findings are consistent with the rapidly emerging research on COVID and mental health and well-being. It is an original contribution to the literature and expands on the role of tele-health and remote support and psychoeducation.

Research Team Response: Thank you for your feedback!

Reviewer 2 Report

Review of Exploring Trauma and Resilience among NYS COVID-19 Pandemic Survivors

It is a very important study to be presented in the field and it has several points of interest to general and specialized readers. However some considerations were addressed to the authors.

1.       Please add subsections to items as this one: 1.1 Exploring Trauma and Resilience among NYS COVID-19 Pandemic Survivors, 1.2 Trauma, 1.3 Limited Treatment Options, Uncertainty Abounded… and so on.

2.       Please add DOI in references, when available.

3.       I personally believe it would be better if authors separate traumas and resilience topics, as it is the title of the study, better. It seemed to me that the topics Psychosocial Interventions, Referrals and Problem Solving are in the topic resilience as it is written in line 462. It’s a matter as noted in item 1 of putting topics as sections, subsections and so on.

4.       Also, it would enrich article if authors in the theme trauma and resilience, could generate a table of any sort of classification. For example, which were the traumas reported and resilience adopted? Or any form of classification of the themes trauma and resilience, because it helps reader to get a general overview of it. But, its up to the authors to check if it is possible.

5.       Also, concerning appendix A and B codes, put it as a citation in the text where it appears first. When reading the article I just paid attention to it when scrolling pages to the end.

6.       I fell a lack of classification aspect of the caller’s data in general. I ask authors, how these written data along the article could help other researchers in terms of decidability aspect? Because the codes generated aren’t that clear in terms of decidability aspect. But check this reviewer suitability to do it and reply back.

7.       I mean which term decisions other important information of the study. For example, Which decisions can be adopted using telehealth? which were the pros and cons of it? how it could be better promoted to generate social adherence? telehealth itself has any intrinsic limitation?

8.       Pick the item 5 answer and do your conclusion section. It will enrich article a lot.

9.       Limitations section address much of what I have presented here as a concern. But no problems. The study have its value even if not initiated under controlled conditions. However, methodology, as I stated need to be very clear and explicit.

10.   As the study is intended to go to special issue “Frontiers in Resilience Psychology” and it has some aspects of cultural and sociocultural factors, it is up to the editor to check article’s suitability and the information contained in the limitations section.

11.   Would be interesting putting in the title information about telehealth tool? Its up to you decide it.

12.   One last point that could enrich much more this research is getting more references in the topic Telehealth. Since it was widely adopted worldwide, recently lots of articles arise describing the theme. It is just a suggestion and not a mandatory comment as the others before.

Author Response

Reviewer Comment #1: Please add subsections to items as this one: 1.1 Exploring Trauma and Resilience among NYS COVID-19 Pandemic Survivors, 1.2 Trauma, 1.3 Limited Treatment Options, Uncertainty Abounded… and so on.

Research Team Response: Thank you for this suggestion that will help us to provide further clarity to the reader. We added the subsections to all items that were missing the notation.

Reviewer Comment #2: Please add DOI in references, when available.

Research Team Response: Thank you for pointing out this oversight – the authors have added DOIs in all references where available.

Reviewer Comment #3: I personally believe it would be better if authors separate traumas and resilience topics, as it is the title of the study, better. It seemed to me that the topics Psychosocial Interventions, Referrals and Problem Solving are in the topic resilience as it is written in line 462. It’s a matter as noted in item 1 of putting topics as section, subsections and so on.

Research Team Response: Thank you for this suggestion. We agree with your rationale and made the categorization more explicit in the Results section with the following statement: “Thus the themes fell into two overarching categories, Trauma and Anxiety, and Resilience (see Figure S3).” The Results section headings and subheadings were also re-arranged to reflect this categorization. We further added a figure to better visually demonstrate where the themes fell in regards to trauma and resilience (addressed in comment #4).

Reviewer Comment #4: Also, it would enrich article if authors in the theme trauma and resilience, could generate a table of any sort of classification. For example, which were the traumas reported and resilience adopted? Or any form of classification of the themes trauma and resilience, because it helps reader to get a general overview of it. But, its up to the authors to check if it is possible.

Research Team Response: Thank you for suggesting this – we agree that it will add to the reader’s understanding of the themes and subthemes. We added a new figure to reflect how the themes were conceptualized based on trauma and resilience. We included the most salient themes in this figure, which were the ones reported in the results section. We added this figure as Figure S3 “Categorization of Trauma and Resilience” and included the following paragraph in the results section to describe this new figure:

“Figure S3 was designed to illustrate the traumatic content that emerged from three parent nodes (i.e., Personal Concerns, Extreme Distress, and Family Concerns), and the resilient content that emerged from three child nodes of the Recommendations parent node (i.e., Psychosocial Interventions, Problem Solving, and Referrals). Not every caller demonstrated both trauma and resilience: most conveyed some form of trauma and/or anxiety, but not all showed resilience as well. Thus, there is no predictive relationship between trauma and resilience implied in this graphic. The graphic is provided to give the reader an idea of how the authors grouped these two constructs according to the qualitative data that emerged.”

Reviewer Comment #5: Also, concerning appendix A and B codes, put it as a citation in the text where it appears first. When reading the article, I just paid attention to it when scrolling pages to the end.

Research Team Response: Thank you for this suggestion. The citation for appendix A – Figure S1 was added to line 598 and the citation for appendix A – Figure S2 was added to line 599.

Reviewer Comment #6: I feel a lack of classification aspect of the caller’s data in general. I ask authors, how these written data along the article could help other researchers in terms of decidability aspect? Because the codes generated aren’t that clear in terms of decidability aspect. But check this reviewer suitability to do it and reply back.

Research Team Response: Thank you for this question. A limitation of this study is that it is a retrospective study utilizing archival data from phone call logs. The initial purpose of these phone logs was to serve as notes for supervision of the doctoral student phone agents. This limitation is addressed in line 638. We aim to add further context to the experience of trauma and resilience in our study but, due to the limitations of our data collection process, do not aim to add to literature on the decidability aspect.

We also added the following sentences to “Study Design” to further address these concerns: “The archival data were collected first as clinical notes, consisting of the nature of the calls and outcomes of the calls, during the agent’s work on the Helpline. After obtaining permission from the Fordham University institutional review board and the NYS Nathan Kline Institute for Psychiatric Research to utilize the clinical notes for the purpose of this retrospective study, the clinical notes were converted to transcripts and coded for themes”. Additionally, the following section, “Sampling Strategy,” further addresses the concerns identified and a sentence as added to provide additional clarification: “All clinical documents submitted were included in the data analytic strategy and any identifiable information (besides gender) that could be traced back to callers was removed to ensure confidentiality. The services provided via the NYS COVID-19 Emotional Support Helpline were anonymous and confidential and the research team cannot trace the data back to specific callers. Additionally, callers’ names were not collected or recorded further protecting callers’ confidentiality.”

Reviewer Comment #7:  I mean which term decisions other important information of the study. For example, which decisions can be adopted using telehealth? Which were the pros and cons of it? How it could be better promoted to generate social adherence? Telehealth itself has any intrinsic limitation?

Research Team Response: Thank you for this feedback. We find that, due to our study having the limitation of being a retrospective study, we did not intend to assess the benefits or disadvantages of telehealth when the data was being collected. Therefore, we feel it is beyond our scope of purpose or experience for us to suggest to others researchers what decisions can be adopted from telehealth. However, we do address in lines 655 – 657 suggestions for education and training of future mental health practitioners to have telehealth integrated into their education. Additionally, lines 643 – 650 address limitations of telehealth in terms of qualitative data analysis.

Reviewer Comment #8: Pick the item 5 answer and do your conclusion section. It will enrich article a lot.

Research Team Response:  We were unclear about how to address this feedback and emailed the editing team for guidance. To attempt to address this in the meantime, we revised the last few sentences in the first paragraph of the discussion section to clarify for the reader as follows:

“Coding revealed nine parent nodes/themes (see Appendix – Figure S1), 39 child nodes/subthemes, and four grandchild nodes/subthemes (see Appendix – Figure S2). From these themes and subthemes emerged ample evidence to suggest that callers experienced symptoms of different posttraumatic stress reactions while also demonstrating resilience with the assistance and support of the agents.”

Reviewer Comment #9: Limitations section address much of what I have presented here as a concern. But no problems. The study has its value even if not initiated under controlled conditions. However, methodology, as I stated, needs to be very clear and explicit.

Research Team Response: We edited the methodology section to be more clear and explicit about the conditions and data collection. See our response to comment #6 for more information.

Reviewer Comment #10: As the study is intended to go to special issue “Frontiers in Resilience Psychology” and it has some aspects of cultural and sociocultural factors, it is up to the editor to check article’s suitability and the information contained in the limitations section.

Research Team Response: Thank you for this feedback. We feel that, although it is a significant limitation that this study does not address cultural or sociocultural aspects within the caller’s lives as much as the authors would have preferred, that this study still fits the special issues intended purpose: to contextualize experiences of resilience. We find our study examines individual resilience from a contextual perspective of limited resources and sociocultural supports within the larger cultural context of New York at the height of the state’s COVID-19 crisis. We find the development and implementation of the hotline at a moment of great urgency was innovative and an attempt to address trauma and foster resilience for community members in real time.

Reviewer Comment #11: Would be interesting putting in the title information about telehealth tool? Its up to you to decide it.

Research Team Response: Thank you for this suggestion. We are not familiar with all of the telehealth platform information that was utilized for the Helpline so we do not feel it is within our scope of knowledge or experience to specifically address the telehealth tool.

Reviewer Comment #12: One last point that could enrich much more of this research is getting more references in the topic Telehealth. Since it was widely adopted worldwide, recently lots of articles arise describing the theme. It is just a suggestion and not a mandatory comment as the others before.

Research Team Response: Thank you for pointing out another way in which we can strengthen and contextualize our examination of Telehealth. While we attempted to include references regarding telehealth interventions prior to and during the pandemic, (e.g., Madigan et al., 2021; Turkington et al., 2020), we have now added the following references on telehealth to the article:

Ibragimov, K., Palma, M., Keane, G. et al. (2022). Shifting to tele-mental health in humanitarian and crisis settings: An evaluation of Medecins Sans Frontieres experience during the COVID-19 pandemic. Confl Health 16(6). https://doi.org/10.1186/s13031-022-00437-1

Reinhardt, I., Gouzoulis-Mayfrank, E., & Zielasek, J. (2019). Use of telepsychiatry in emergency and crisis intervention: Current evidence. Curr Psychiatry Rep 21(63). https://doi.org/10.1007/s11920-019-1054-8

Brenna, C.T., Links, P.S., Tran, M.M., Sinyor, M., Heisel, M.J., & Hatcher, S. (2021). Innovations in suicide assessment and prevention during pandemics. Public Health Research & Practice, 31(3). DOI: 10.17061/phrp3132111. PMID: 3449407

Reviewer 3 Report

This manuscript described the experiences of a real world happened in the COVID-19 pandemic. Although it was not a well-designed study, the records and analysis of the telephone help-seeking contents were important for psychological professionals to develop client-helping strategies.

My only suggestion is that the authors need to describe the aim of this study clearly in Abstract.

Author Response

Reviewer Comment #1: My only suggestion is that the authors need to describe the aim of this study clearly in the Abstract.

Research Team Response: Thank you for this feedback. The writers added the following sentence to the abstract to more clearly describe the aim of this study: “The purpose of this retrospective qualitative study was to explore the themes that emerged from calls to give voice to the trauma callers were reporting during the early phases of the pandemic, and the resilience they demonstrated as they engaged with the Helpline.”

Reviewer 4 Report

This report is a qualitative study of doctoral students providing phone support for callers from New York State community during the early parts of the pandemic from May to August 2020. The study examines the common themes expressed by callers, the kinds of anxieties and traumas expressed and their resilience. As a result, this research contributes to our understanding of how telehealth may contribute to managing well-being during natural crises. The following should be considered:

1.       The introduction is too long and does not directly lead to the purposes of the current study.

2.       The discussion should compare these results to other studies that have been done on individuals’ response and resilience during the pandemic.

3.       The discussion could reference the many resources that have been disseminated about providing telehealth or tele-mental health during pandemics or natural crises. For example,

Innovations in suicide assessment and prevention during pandemics.

Brenna CT, Links PS, Tran MM, Sinyor M, Heisel MJ, Hatcher S.

Public Health Res Pract. 2021 Sep 8;31(3):3132111. doi: 10.17061/phrp3132111.

PMID: 34494071

Author Response

Reviewer Comment #1: The introduction is too long and does not directly lead to the purposes of the current study.

Research Team Response: Thank you for this feedback. The following sentence was added to the introduction to further clarify the purpose of the study: “Thus, this qualitative investigation aimed to retrospectively explore the traumas NYS COVID-19 Emotional Support Helpline callers were reporting and the resilience they demonstrated through the use of the service.

Reviewer Comment #2: The discussion should compare these results to other studies that have been done on individuals’ response and resilience during the pandemic.

Research Team Response: Thank you. We added the following paragraph to the discussion to address this feedback.

“Other studies examining the impact of COVID-19 on communities across the globe discovered similar depictions of trauma and anxiety to those found in the current study such as Mboua et al.’s study that reported elevated symptoms for Cameroonian citizens during the pandemic [27]. This study’s findings also parallel others examining resilience amidst the COVID-19 pandemic that suggest that most individuals have the capacity for resilience, particularly when able to connect with others [28, 29]. beyond connecting and sharing their troubles with agents, this study uniquely demonstrated the specific interventions (providing psychosocial support, referrals, and by engaging in problem solving) that engaged callers’ resilience and helped them cope with their circumstances.” The Mboua, Cenat, and PeConga references reflect other studies that have been conducted on individuals’ responses and we hope that this satisfies this request.

Reviewer Comment #3: The discussion could reference the many resources that have been disseminated about providing telehealth or tele-mental health during pandemics or natural crises. For example,

Innovations in suicide assessment and prevention during pandemics.

Brenna CT, Links PS, Tran MM, Sinyor, M, Heisel MJ, Hatcher S.

Public Health Res Pract. 2021 Sep 8; 31(3):3132111. Doi: 10.17061/phrp3132111.

PMID: 34494071

Research Team Response: Thank you for this suggestion and for the example reference. The following citations were added to the manuscript:

Ibragimov, K., Palma, M., Keane, G. et al. (2022). Shifting to tele-mental health in humanitarian and crisis settings: An evaluation of Medecins Sans Frontieres experience during the COVID-19 pandemic. Confl Health 16(6). https://doi.org/10.1186/s13031-022-00437-1

Reinhardt, I., Gouzoulis-Mayfrank, E., & Zielasek, J. (2019). Use of telepsychiatry in emergency and crisis intervention: Current evidence. Curr Psychiatry Rep 21(63). https://doi.org/10.1007/s11920-019-1054-8

Brenna, C.T., Links, P.S., Tran, M.M., Sinyor, M., Heisel, M.J., & Hatcher, S. (2021). Innovations in suicide assessment and prevention during pandemics. Public Health Research & Practice, 31(3). DOI: 10.17061/phrp3132111. PMID: 3449407

Round 2

Reviewer 2 Report

Thank for the improvements performed. I have no more issues to address.